# High-Throughput Sequencing Reveals the Mycoviral Diversity of the Pathogenic Grape Fungus *Penicillium astrolabium* During Postharvest

**DOI:** 10.3390/v17081053

**Published:** 2025-07-28

**Authors:** Rui Wang, Guoqin Wen, Xiaohong Liu, Yingqing Luo, Yanhua Chang, Guoqi Li, Tingfu Zhang

**Affiliations:** 1Key Laboratory of Southwest China Wildlife Resource Conservation (Ministry of Education), College of Life Science, China West Normal University, Nanchong 637000, China; 15183866305@163.com (R.W.); gqin0817@126.com (G.W.); liuxiaohong13330761109@cwnu.edu.cn (X.L.); luoyingqing5@126.com (Y.L.); yanhuachang2022@163.com (Y.C.); 2Key Laboratory of Pathogenic, Fungi and Mycotoxins of Fujian Province, School of Life Sciences, Fujian Agriculture and Forestry University, Fuzhou 350002, China; liguoqi0927@163.com

**Keywords:** grape, *Penicillium astrolabium*, mycoviral diversity, high-throughput sequencing, biocontrol

## Abstract

*Penicillium astrolabium* is a primary pathogenic fungus that causes grape blue mold during postharvest, leading to substantial losses in the grape industry. Nevertheless, hypovirulence-associated mycoviruses can attenuate the virulence of postharvest grape-rot pathogens, thereby offering a promising biocontrol tool. Characterizing the mycovirus repertoire of *P. astrolabium* is imperative for grape protection, yet remains largely unexplored. Here, we screened six strains harboring viruses in 13 *P. astrolabium* isolates from rotted grapes. Using high-throughput sequencing, four novel dsRNA viruses and two +ssRNA viruses were identified from the six *P. astrolabium* strains. The dsRNA viruses belonged to two families—*Chrysoviridae* and *Partitiviridae*—and were designated to Penicillium astrolabium chrysovirus 1 (PaCV1), Penicillum astrolabium partitivirus 1′ (PaPV1′), Penicillum astrolabium partitivirus 2 (PaPV2), and Penicillum astrolabium partitivirus 3 (PaPV3). For the +ssRNA viruses, one was clustered into the *Alphaflexiviridae* family, while the other one was clustered into the *Narnaviridae* family. The two +ssRNA viruses were named Penicillium astrolabium alphaflexivirus 1 (PaAFV1) and Penicillium astrolabium narnavirus 1 (PaNV1), respectively. Moreover, several viral genomic contigs with non-overlapping and discontinuous sequences were identified in this study, which were probably representatives of five viruses from four families, including *Discoviridae*, *Peribunyaviridae*, *Botourmiaviridae*, and *Picobirnaviridae*. Taken together, our findings could expand the diversity of mycoviruses, advance the understanding of mycovirus evolution in *P. astrolabium*, and provide both potential biocontrol resources and a research system for dissecting virus–fungus–plant interactions.

## 1. Introduction

Grapes (*Vitis vinifera* L.) are one of the most important fresh fruits worldwide. They are often infected by diverse pathogens (e.g., fungi, bacteria, and viruses) throughout their production and marketing, resulting in severe yield and quality losses. Even under cold storage conditions, grapes remain vulnerable to postharvest phytopathogens, including blue mold, gray mold, and rhizopus rot, with documented losses approximating 39% of the total production yield and 30% of the market value [1]. Blue mold caused by *Penicillum* spp. is the main disease during the storage, transportation, and sales of grapes, which seriously reduces their edible and commercial value. However, studies on the *Penicillium* pathogens of grapes are very limited according to the published literature. Recently, *Penicillium olsonii* has been reported as one of the major fungal pathogens of postharvest grapes [2]. It was reported that the morphological characteristics of *P. astrolabium* were similar to those of *P. olsonii*, and symptoms caused by these pathogens were also similar in grapes, thereby complicating the differentiation of two postharvest pathogens in grape production [3]. Nevertheless, we found that *P. astrolabium* was more prevalent than *P. olsonii* in rotten grapes, suggesting that *P. astrolabium* is one of the predominant pathogens causing postharvest blue mold in grapes. At present, the effective control of *Penicillium*-induced postharvest diseases is an urgent priority for the grape industry. Although chemical control can effectively suppress the diseases, some fungi (e.g., *Botrytis cinerea*) readily evolve highly resistant strains, causing a progressive decline in efficacy [4]. Additionally, chemical control methods are not conducive to environmental protection, and the presence of residual drugs can harm the health of humans and animals. However, biocontrol is an effective method to control diseases by using the interactions between organisms. It has only a limited impact on ecological homeostasis and poses little risk of environmental pollution. Therefore, developing potent biocontrol strategies against postharvest diseases in grapes has become a pivotal research priority.

Mycoviruses are a type of virus that can infect fungi. According to previous reports, approximately 30–80% of fungi in nature can be infected by mycoviruses [5]. A study relating to the viral infection of fungi was first reported in 1962 in cultivated mushrooms [6]. Subsequently, more and more viruses were gradually discovered in various fungi. Although the majority of mycoviruses persist asymptomatically within fungal hosts without changing phenotypic traits, a minority can markedly alter the virulence and morphology of their hosts [7]. The most common manifestations include reduced host growth rate, abnormal pigment deposition, and decreased spore production; the resulting attenuation of host pathogenicity is termed hypovirulence [8]. Previous studies demonstrate that certain mycoviruses have been successfully harnessed for plant disease biocontrol applications. For example, Cryphonectria hypovirus 1 (CHV1) has been used to control chestnut blight caused by *Cryphonectria parasitica* in Europe [9], and Rosellinia necatrix megabirnavirus 1 was also applied to control the apple white root rot disease [10,11,12]. Notably, recent studies have revealed that mycoviruses can transform pathogenic fungi into beneficial endophytes and activate plant immunity [13,14]. Sclerotinia sclerotiorum hypovirulence-associated DNA virus 1 (SsHADV-1) isolated from *Sclerotinia sclerotiorum* could protect oilseed rape from damage caused by a highly virulent strain of *S. sclerotiorum* [15]. Despite the growing number of identified mycoviruses, there are some shortcomings such as low throughput, narrow coverage, incomplete viral genome sequences, and unknown viral functions, which limits the discovery of mycoviral resources due to insufficient knowledge regarding their comprehensiveness and completeness.

With the development of high-throughput sequencing technology and the application of RNA-Seq in virus research, an increasing number of novel mycoviruses are being discovered and they are widely distributed across diverse fungal species. To date, the International Committee on Taxonomy of Viruses (ICTV) has recorded hundreds of mycoviruses, which are mainly distributed in the form of ascomycetes and basidiomycetes [16,17,18]. The ICTV reported 41 mycovirus families, as well as their genomic forms including double-stranded RNA (dsRNA), positive single-stranded RNA (+ssRNA), negative single-stranded RNA (-ssRNA), reverse-transcribing single-stranded RNA (ssRNA-RT), and single-stranded DNA (ssDNA). According to previous reports, most of the mycoviruses possess dsRNA viruses (including the dsRNA genome), and were categorized into 16 different families, as well as an unclassified genus named *Rhizidiovirus*. The ssRNA viruses are classified into 21 families, including 13 family members with the +ssRNA genome and 4 family members with the -ssRNA genome. The remaining four families covered viruses of the +ssRNA or -ssRNA genotypes. Moreover, ssDNA viruses have only one family—*Genomoviridae* [11]. With the development of bioinformatics, RNA-Seq has not only been an important method for transcriptome analysis in different organisms [19] but has also become an important tool for new virus discovery. For instance, Gilbert et al. (2019) identified 59 viruses from 44 different fungi; among the identified viruses, 88% were determined to be new species and 68% were the first viruses described from the fungal species. Among them, 57 viruses belonged to 12 families, including *Partitiviridae*, *Totiviridae*, *Fusariviridae*, *Mitovirus*, *Hypoviridae*, *Endornaviridae*, *Ambiguiviridae*, *Alternaviridae*, *Chrysoviridae*, *Yadokariviridae*, *Tobamo-like virus*, and *Ourmia-like virus*, and the remaining 2 viruses were unclassified [20]. Thus, high-throughput sequencing provides an opportunity to enrich the diversity of mycoviruses.

Although several mycoviruses had previously been reported in different *Penicillium* species, it was not until 2003 that the first viral genome *Penicillum stoloniferum* virus S was sequenced in *Penicillium* species [21]. Up to now, more than 40 viruses have been found in *Penicillium* spp. and are distributed across a total of 11 families. From previous reports, it was found that there were at least 20 *Penicillium* hosts that can be infected by mycoviruses, including *P. digitatum*, *P. janczewskii*, *P. aurantiogriseum*, etc. [22,23,24,25,26,27,28,29,30]. Among these viruses harbored in *Penicillium* spp., about 75% of them were dsRNA viruses that could further be classified into the four families—*Chrysoviridae*, *Partitiviridae*, *Totiviridae*, and *Polymycoviridae*—while the remaining small number of viruses were ssRNA viruses belonging to six families. In 2022, Fu et al. reported a homologous virus SsHADV1_PO with *Sclerotinia sclerotiorum* hypovirulence-associated DNA virus 1 (SsHADV-1) from *P. olsonii* [31]. Importantly, the classification of some recorded mycoviruses has been revised by the ICTV. For example, Jiang and Ghabrial first published the whole-genome sequence of the *P. chrysogenum* virus (PcV) and suggested that PcV was subordinated to the genus *Chrysovirus of Chrysovidae* rather than the *Partitiviridae* family [32]. By 2018, the genus *Chrysovirus* was further revised to *Alphachrysovirus* and *Betachrysovirus* [16]. Although the taxonomy of mycoviruses is constantly being updated, some mycoviruses are still unclassified due to the lack of other sister clade members. To date, several mycoviruses with biocontrol potential have been reported in *Penicillium* species, such as Penicillium digitatum virus 1 (PdV1), Penicillium digitatum polymycovirus 1 (PdPMV1), and Penicillium digitatum Narna-like virus 1 (PdNLV1) [23]. However, due to the scarcity of viral resources discovered in *P. astrolabium*, it remains unknown whether any such viruses possess applied value for biocontrol. Therefore, we employed high-throughput sequencing technology to identify the viruses harbored in *P. astrolabium* isolated from grapes. Our study could enrich the diversity of virus species harbored in *Penicillium* spp. and provide some potential mycovirus resources to be used for virus study and biocontrol.

## 2. Materials and Methods

### 2.1. Fungal Isolates and Culture Conditions

The fungal isolates WHG3-3, WHG6-2, WHG8, WHG9, WHG10, and WHG11 were originally isolated from the mildew fruits of postharvest grape in Wuhan city, Hubei province, China. They were maintained on potato dextrose agar (PDA) at 28 °C to produce conidia. Mycelium for DNA or RNA extraction were cultured in potato dextrose broth (PDB; PDA without agar) by conidial germination on a rotary shaker with 180 rpm at 28 °C for 5 d.

### 2.2. Genomic DNA Extraction and Isolates Harboring Mycoviruses Screening

Total genomic DNA was extracted from ground mycelia powder in liquid nitrogen using the Biospin Fungus Genomic DNA Extraction Kit (Hangzhou Bioer Tec. Co., Ltd., Hangzhou, China), according to the manufacturer’s protocol. After culturing all isolated strains in PDB, the mycelia were collected and the total genome was subsequently extracted for agarose gel electrophoresis detection to assess isolates harboring viruses. Based on the electrophoretic map, we screened for the fungal strains harboring viruses by checking whether the fungal total genomic DNA preparations contained additional nucleic acid bands corresponding to dsRNA elements. Mycelia harboring mycoviruses were frozen with liquid nitrogen and ground to a fine powder. The dsRNA was extracted with phenol-chloroform, before being isolated using CF-11 cellulose (Sigma, St Louis, MO, USA) adsorption chromatography [33], followed by separation on 1% agarose gels and staining with ethidium bromide.

### 2.3. RNA Extraction, Sequencing Library Preparation, and RNA-Seq

Total RNA was extracted from the mycelia using TRIzol^®^ Reagent, according the manufacturer’s instructions (Invitrogen, Carlsbard, CA, USA), and genomic DNA was digested using DNase I (Takara, Dalian, China). The integrity and purity of the total RNA was determined using the 2100 Bioanalyser (Agilent Technologies, Inc., Santa Clara, CA, USA) and quantified using the ND-2000 (NanoDrop Thermo Scientific, Wilmington, DE, USA). Then, the high-quality nucleic acid samples (i.e., DNA OD260/280 = 1.8~2.0, RNA OD260/280 = 1.8~2.2, OD260/230 ≥ 2.0, RIN ≥ 8.0, and 28S:18S ≥ 1.0) were used to construct an RNA sequencing library.

RNA-Seq for transcriptome-containing virome libraries was prepared using the Illumina TruSeqTM RNA sample preparation Kit (San Diego, CA, USA). For transcriptome sequencing, Poly(A) mRNA was, respectively, purified from 3 μg total RNA through oligo-dT-attached magnetic beads and then fragmented using a fragmentation buffer for isolates harboring mycoviruses. Then, taking the fragmentation of RNA as a template, double-stranded cDNA was, respectively, synthesized using the SuperScript double-stranded cDNA synthesis kit (Invitrogen, CA, USA) with random hexamer primers (Illumina, San Diego, CA, USA). The ends of synthesized cDNA were repaired and phosphorylated, and an ‘A’ base was added to the 3′-end, according to the Illumina library construction protocol. Libraries were selected for cDNA target fragments size of 200–300 bp on 2% Low Range Ultra Agarose followed by PCR, which was amplified using Phusion DNA polymerase (New England Biolabs, Boston, MA, USA) for 15 PCR cycles. After quantification using TBS380, the RNA-Seq libraries were sequenced using paired-end sequencing with a 150 bp read length on an Illumina Hiseq X platform (Illumina, San Diego, CA, USA).

### 2.4. High-Throughput Sequencing and Data Processing and Analysis

Library sequencing was carried out using the Illumina Hiseq X platform from Majorbio Co., Ltd., (Shanghai, China), and every sample generated raw data of more than 6 G, corresponding to approximately 200× coverage. The computational steps for identifying viral sequences started with reads from *P. astrolabium* RNA-Seq raw data. Adapters and low-quality reads were removed using Trimmomatic software (version 0.36) to obtain the clean data [34]. The clean reads were mapped to the reference genome of *P. astrolabium* using HISAT2 (version 2.1.0) [35]. The remaining reads that were unmapped to the reference genome of host fungi were extracted using SAMtools (version 1.9) [36]. Subsequently, contigs were generated by *de novo* assembly using Trinity (version 2.5.0) and SPAdes (version 3.11.1) [37]. All spliced contigs were removed from redundant sequences using CD-HIT [38], and the longest contigs among these sequences with an identity of more than 90% were selected for subsequent analysis. The non-redundant splicing contigs were aligned with the NCBI nr database using DIAMOND BLASTX (version 0.9.10) [39], and the E-value was set at the default value of 10^−5^. Finally, contig sequences that were aligned to known viral information were further extracted.

### 2.5. Analysis of Conserved Domains of Viral RdRps and Phylogenetic Tree Construction

The obtained nucleotide sequences were subjected to the BLASTx program on the NCBI website to analyze the sequence similarity. The ORFs encoding RdRp were predicted with ORF Finder on the NCBI website, before being deduced and translated by DNAMAN (version 7.0) with the default parameters [40]. The multiple sequence alignment of every family virus, RdRps, and the related orthologs was generated using Clustal W (version 2.1). The conserved domains were analyzed and phylogenetic trees based on RdRps were constructed using the neighbor-joining method, as described previously, with a bootstrapping analysis of 1000 replicates using MEGA (version 5.0) [41].

## 3. Results

### 3.1. Screening of P. astrolabium Strains Harboring Viruses

*Penicillium* spp. is one of the main pathogens causing grape rot during postharvest. In total, 25 strains were isolated on decayed grapes from Wuhan City, China. In total, 13 strains of these isolates were preliminarily identified as *P. astrolabium* according to ITS sequencing and BLAST analysis. Then, their total genomic DNA was, respectively, extracted from ground mycelia powder, and the electrophoretic result showed that six strains including WHG3-3, WHG6-2, WHG8, WHG9, WHG10, and WHG11 contained several additional nucleic acid segments of their genomes. These were probably the genomic RNA of mycoviruses (Figure 1A). The dsRNA was further extracted using the phenol-chloroform method and isolated using CF-11 cellulose chromatography. The electrophoresis map of the dsRNA from the six strains indicated that there were seven bands in the WHG8 strain and at least three obvious bands in the other five strains (Figure 1B–D). According to the electrophoresis analysis of dsRNA, the virus carrying rate was approximately 46% in the *P. astrolabium* strains. The results demonstrated that a total of six *P. astrolabium* strains harboring viruses were initially identified and screened from rotten grapes.

### 3.2. Sequence Analysis of the Chrysoviridae Virus

*Chrysoviridae* is a family including two genera virus members with multi-segmented, double-stranded RNA genomes. The dsRNA segments are individually encapsidated in separate particles with a dsRNA length from 2.2 to 4.0 kb [18]. In this study, there were 13 different contigs of the *Chrysoviridae* family, with sizes of more than 500 bp, that were initially assembled from reads unmapped to the reference genome of the host *P. astrolabium* in the WHG8 isolate. After alignment and splicing by DNAMAN, four longer contigs (corresponding to four dsRNAs) were assembled, with lengths of 3709 bp, 2812 bp, 3065 bp, and 2684 bp, coding 1125 amino acids (aas), 765 aas, 819 aas, and 686 aas, respectively (Figure 2A). The amino acid sequence deduced from dsRNA1 exhibited an 83.48% identity to the RNA-dependent RNA polymerase (RdRp) of Penicillium janczewskii chrysovirus 1 (PjCV1) according to BLASTX analysis. There were eight conserved motifs in the amino acid sequence deduced from dsRNA1 and the RdRps selected from the dsRNA viruses of *Chrysoviridae* (Figure 2B). The identity between dsRNA2 coding putative proteins and PjCV1 coat protein (CP) was 73.64%; the identity between dsRNA3/4 coding proteins and PjCV1 P3/4 was 47.27% and 77.44%, respectively. On these grounds, the virus in *P. astrolabium* was named as Penicillium astrolabium chrysovirus 1 (PaCV1). Phylogenetic analyses based on the RdRp sequences of members selected from *Chrysoviridae* family showed that the RdRp of PaCV1 strongly resembled PjCV1 RdRp in a clade within the genus *Betachrysovirus*, which indicated that PaCV1 was a new member of the *Betachrysovirus* genus (Figure 2C).

### 3.3. Sequence Analysis of the Partitiviridae Virus

*Partitiviridae* includes five genera and possesses two essential genome segments—dsRNA1 and dsRNA2—each containing one ORF on the positive-strand RNA. One segment (dsRNA1) encodes RdRp, and the other segment (dsRNA2) encodes CP. The linear segments are 1.4 to 2.4 kb in length [42]. *P. astrolabium* RNA-Seq analysis revealed 12 partitivirus-related contigs with a length of more than 500 bp in all six strains. Five longer contigs of novel viruses were further assembled by DNAMAN. The putative proteins deduced from contigs 3 (1752 bp, coding 486 aa) and 4 (1737 bp, coding 274 aa) shared 99.26% (RdRp) and 98.52% (CP) identity with the Penicillium stoloniferum virus F (PsV-F) according to BLASTx analysis, respectively. Therefore, this virus has been tentatively named Penicillum astrolabium partitivirus 1′ (PaPV1′), which was distinguished from Penicillium aurantiogriseum partitivirus 1 (PaPV1). BLASTX searches of the deduced amino acid sequence of contigs 1 (1895 bp, coding 538 aa) and 5 (1730 bp, coding 485 aa) revealed homology (86.57% and 52.73% identity) with the RdRp and CP of PaPV1, respectively. As such, this virus was temporarily designated as Penicillum astrolabium partitivirus 2 (PaPV2). Contig 2 with a 1876 bp encoding a 573 aa putative protein had a 63.41% identity with Diplodia seriata partitivirus 1 RdRp. Although the CP coding information for this virus has not yet been assembled from RNA-Seq data, it was provisionally proposed that this would be called Penicillum astrolabium partitivirus 3 (PaPV3). The genomic structures of PaPV1′ and PaPV2, respectively, consisted of two monocistronic segments, while PaPV3 contained only one monocistronic structure (Figure 3A). The alignment of the RdRps from eight partitiviruses, including PaPV1′, PaPV2, and PaPV3, revealed the presence of six conserved domains, which were the same as other known partitiviruses (Figure 3B). The RdRps of PaPV1′, PaPV2, and PaPV3 were used for phylogenetic analysis with the RdRps of other members belonging to different genera in the *Partitiviridae* family. The results indicated that PaPV1′and PaPV2 were clustered into the same branch as members of the *Gammapartitivirus* family; therefore, we recommended the two viruses in the *Gammapartitivirus* genus to be new viral species. Meanwhile, PaPV3 was clustered in an unclassified *Partitivirus* with other virus members (Figure 3C). Thus, it was considered a new member of the unclassified virus in the *Partitiviridae* family.

### 3.4. Sequence Analysis of the Alphaflexiviridae Virus

*Alphaflexiviridae* includes seven genera whose genome consists of a single-stranded and positive-sense RNA of 5.4 to 9 kb in length [43]. RNA-Seq analysis revealed that a total of 19 virus contigs with a length of more than 500 bp were assembled in the WHG3-3, WHG8, WHG9, WHG10, and WHG11 strains. A sense single-stranded RNA, with a length of 5213 bp, was further assembled by DNAMAN 7.0, including three ORFs that coded the replicase (Rep), CP, and Triple Gene Block 1 (TGB1), respectively. The results of BLASTX analysis demonstrated that the identities of the three putative proteins were 99.12%, 98.12%, and 98.18% with the Rep, CP, and TGB1 of Narcissus mosaic virus (NMV), respectively. Therefore, the virus was designated as Penicillium astrolabium alphaflexivirus 1 (PaAFV1). The genomic structure of PaAFV1 showed that Rep, TGB, and CP were separated by non-coding sequences (Figure 4A). There were eight conserved motifs in the putative protein encoded by PaAFV1 ORF1, and these motifs existed in the replicases of other members in the *Alphaflexivirida* family (Figure 4B). The Rep sequences of PaAFV1 and other viruses from different genera of the *Alphaflexiviridae* family were selected for phylogenetic analysis. The results showed that PaAFV1 and Narcissus mosaic virus were clustered into the same branch, with a support coefficient of 100 in the genus *Potexvirus* (Figure 4C). Therefore, PaAFV1 should be a new viral member within the *Potexvirus* genus.

### 3.5. Sequence Analysis of the Narnaviridae Virus

Viruses in the *Narnaviridae* family consist of a single and non-encapsidated positive-stranded RNA of 2.3 to 2.9 kb, encoding a single protein of 80 to 104 kDa with amino acid sequence motifs characteristic of an RdRp (https://ictv.global/report_9th/RNApos/Narnaviridae, accessed on 26 August 2024). In total, 24 contigs with a length of more than 500 bp were assembled from the RNA-Seq data of all six *P. astrolabium* strains. They were further assembled to form three longer contigs with lengths of 2522 bp, 2512 bp, and 2137 bp, which encoded proteins containing 768 aas, 779 aas, and 641 aas, respectively. The BLASTx analysis showed that the three putative proteins shared identities of 73.99%, 74.74%, and 66.78% with RNA1 RdRp, and RNA2 RdRps, as well as a hypothetical protein of Oidiodendron maius splipalmivirus 1. Therefore, we speculate that the three contigs represent the three genome segments of Penicillium astrolabium narnavirus 1 [44]. According to the nomenclature established for Oidiodendron maius splipalmivirus 1, we temporarily suggested that the virus name was Penicillium astrolabium narnavirus 1, with two RNAs encoding RdRp. The genomic structure is shown in Figure 5A. The RdRps of PaNV1 RNA1 and RNA2 were used for a phylogenetic analysis with other RdRps of different viruses in the *Narnaviridae* family. The result showed that both PaNV1 RNA1 and PaNV1 RNA2 belonged to unclassified narnaviruses (Figure 5B). Therefore, the Penicillium astrolabium narnavirus 1 was considered a novel and unclassified virus in the *Narnaviridae* family.

## 4. Discussion

Mycoviruses are a type of virus that can infect fungi in the growth and multiplying stages; they are widely distributed in different fungal groups [23]. Due to their hypovirulence, mycoviruses can reduce the virulence of their fungal hosts and thus hold potential for biocontrol applications. To the best of our knowledge, approximately 40 species of *Penicillium* have been found to harbor viruses [22,23,24,25,26,27,28,29,30]. Niu et al. isolated and characterized a mycovirus from *P. digitatum* that contributed to the hypovirulence phenotypes of the host strains [26]. Afterwards, two mycoviruses named PdPMV1 (Penicillium digitatum polymycovirus 1) and PdNLV1 (Penicillium digitatum Narna-like virus 1) were identified in *P. digitatum*, and they could reduce the resistance of their hosts to prochloraz, thereby improving the efficiency of pesticide use [23]. However, to date, mycoviruses that can be used for the prevention and control of diseases caused by *Pencillium* spp. remain very limited. Therefore, it is necessary to explore more mycovirus resources for application in the biocontrol of fungal diseases, particularly postharvest fungal diseases of fruits and vegetables. Currently, high-throughput sequencing is widely applied to determine the sequence of biomacromolecule DNA or RNA. This technology has provided an important basis for research on species evolution. In terms of mycoviruses, high-throughput sequencing could analyze their species characteristics and evolutionary rules by obtaining specific virus sequences. Gilbert et al. identified 59 viruses from 44 different fungi, and these viruses were classified into 12 families [20]. Notably, a large number of novel mycoviruses have been discovered using this technique, which has significantly expanded our understanding of virus diversity [44,45].

In this study, six *P. astrolabium* strains harboring viruses were screened in 25 *Pencillium* strains isolated from postharvest grapes. These strains harboring viruses were identified using RNA-Seq to explore virus diversity. The results showed that each *P. astrolabium* strain was infected by 5–9 viruses (Appendix A). These viruses belonged to the *Chrysoviridae*, *Partitiviridae*, *Narnaviridae*, and *Alphaflexiviridae* families. Members of the *Chrysoviridae* and *Partitiviridae* family were common dsRNA viruses, and the former had multipartite genomes comprising three to seven linear monocistronic dsRNA segments [16]. PaCV1 was the same as most mycoviruses belonging to the genus *Betachrysovirus* in the *Chrysoviridae* family, possessing four monocistronic segment genomes. Its dsRNA1 and dsRNA2 encode RdRp and CP, respectively, and the remaining two RNAs encode proteins p3 and p4, for which their functions are currently unknown. It was reported that chrysoviruses infected many fungi, such as *Fusarium graminearum*, *Magnaporthe oryzae*, and *P. crustosum*, which caused phenotypic changes and reduced host virulence or fungicide resistance [46,47,48]. Whether the PaCV1 identified in this study can affect host phenotypes or not needs to be further studied. With regard to *Partitiviridae* viruses, members of all five genera possessed two essential genome segments—dsRNA1 and dsRNA2. The large segment (dsRNA1) was a positive-stranded RNA encoding RdRp and the other segment (dsRNA2) encodes CP. Some partitiviruses had an additional (satellite or defective) dsRNA element [42]. The dsRNA electrophoretic map revealed that partitiviruses harbored in six *P. astrolabium* strains had three segments of genomic dsRNA (Figure 1B). According to the literature, RNA3 with a size of 1.3 kb may be a satellite RNA or a genomic segment encoding another CP in a few partitiviruses [42,49]. However, the sequences with homology to satellite RNA or another CP-encoded segment of the comparable molecular weight have not yet been obtained from RNA-Seq data. Therefore, it is necessary to further sequence for the RNA3 band by cloning or directly sequencing via high-throughput sequencing, which can obtain the complete genome sequences of these partitiviruses including the three dsRNA segments. Among the five genera of the *Partitiviridae* family, gammapartitiviruses infected only filamentous fungi. Although partitivirus infections were largely symptomless, a few partitiviruses have been reported to attenuate the pathogenicity of host fungi. For example, Sclerotinia sclerotiorum partitivirus 1 (SsPV1) could cause hypovirulence in *Sclerotinia* spp. and *B. cinerea* [50]. Whether these partitiviruses in *P. astrolabium* affect the pathogenicity of their hosts remains to be determined. Collectively, we discovered four novel dsRNA viruses, including one chrysovirus (PaCV1) and three partitiviruses (PaPV1′, PaPV2, and PaPV3), which enriched the diversity of dsRNA mycoviruses and provided potential microbial resources for biocontrol.

Although the genomes of most discovered mycoviruses were dsRNA [51], more and more studies have revealed that ssRNA viruses are also abundant in fungi species. We found three viruses with a +ssRNA genome in *P. astrolabium*, including one virus of the *Narnaviridae* family and one of the *Alphaflexiviridae* family (Appendix A). The structure of the *Narnaviridae* family viruses is the simplest of the known RNA viruses, consisting of an +ssRNA that may be as small as 2.3 kb and encoding only an RdRp used for their own replication (https://ictv.global/report_9th/RNApos/Narnaviridae). However, a narnavirus found in our study contained two RdRps with a lower identity (only 11.68%), which was the same as the Oidiodendron maius splipalmivirus 1 [44]; no obvious conserved motifs existed in other known homologous proteins. This characteristic was inconsistent with typical *Narnaviridae* family viruses. Therefore, we speculate that this may be an evolutionary trajectory of these viruses. Although *Alphaflexiviruses* infected both plants and plant-infecting fungi [43], they were more common in infecting plants. Considerable numbers of plant viruses and mycoviruses shared similar genome characteristics, and some of them were phylogenetically closely related [52,53]. For instance, in the *Alphaflexiviridae* family, mycoviruses belonging to *Botrexvirus* were strikingly similar to plant viruses of the genus *Potexvirus* [54,55,56]. The PaAFV1 identified in this study was classified into the *Potexvirus* family from the fungi *P. astrolabium*. It was reported that fungal strains isolated from plants in fields were discovered to carry plant viruses. Moreover, artificial virus inoculations in the laboratory demonstrated the compatibility of fungi as hosts of plant viruses and vice versa [57,58]. For example, the plant virus named cucumber mosaic virus (CMV) could infect the phytopathogenic fungus *Rhizoctonia solani* across the kingdom and transmit the virus to virus-free plants and fungi, achieving the two-way transmission of plant viruses between plants and fungi [58]. Although viruses co-evolve with their host cells and each virus can only be maintained in specific host cells, the above phenomenon showed that many plant and fungi viruses were taxonomically related and displayed host interchangeability [17]. Therefore, we speculate that PaAFV1 may be transferred from the grapes to the virus host *P. astrolabium*.

In addition, some viral contigs of incomplete genomes were also obtained from the RNA-Seq data in this study, representing three viruses with a genomic structure of -ssRNA, one with a genomic structure of +ssRNA, and a virus with a genomic structure of dsRNA (Appendix A). A total of seven contig sequences of the *Discoviridae* family were assembled in the strains WHG6-2. Three non-overlapping and discontinuous sequences of the seven contigs (1540 nt, 829 nt, and 532 nt) had 17.39%, 9.30%, and 6.48% identities with Penicillium discovirus (PdV) RdRps according to BLASTX, respectively. It is likely that the complete sequence would be homologous to the RdRp of PdV. Of the seven contigs, the two with lengths of 1166 nt and 1207 nt had 83.98% and 65.27% identities with the CP and NS1 of PdV, respectively. The results demonstrated that the five contig sequences may be derived from a virus homologous to PdV. The other two contigs (1758 nt and 1141 nt) had 48.61% and 61.65% identities to Penicillium roseopurpureum negative ssRNA virus 1 (PrNssV1) CP and nonstructural protein 1 (NP1), respectively. However, no contig sequence that was homologous to the PrNssV1 RdRp was detected. Analogously, there were also three non-overlapping and discontinuous contigs that were homologous to a virus of the *Peribunyaviridae* family in strain WHG6-2. All three contigs (2580 nt, 2022 nt, and 1869 nt), respectively, had 46.79%, 39.29%, and 38.26% identities with Guyuan tick virus 1 *RdRp* (6540 nt), assuming that the three contigs were a part of a complete virus sequence that was homologous to the RdRp of Guyuan tick virus 1. In this study, two non-overlapping and discontinuous contigs (1265 nt and 979 nt) derived from the *Botourmiaviridae* +ssRNA virus were assembled from WHG9 RNA-Seq data. They had 48.47% and 64.86% identities with the RdRp sequence of Tongren Botou tick virus 1 according to BLASTX, respectively. Only a non-overlapping and discontinuous contig (1360 nt) derived from the *Picobirnaviridae* dsRNA virus was assembled, which shared 75% identity with the RdRp of the Lysoka partiti-like virus. These viruses have not yet acquired complete genome segments or the RdRps coding frame, but to some extent, they represented five different novel viruses from the *Discoviridae*, *Peribunyaviridae*, *Botourmiaviridae*, and *Picobirnaviridae* families. As such, the missing sequence fragments will be supplemented to further enrich the mycovirus diversity of *P. astrolabium* in subsequent research work.

In this study, six novel viruses were identified from the postharvest grape pathogenic fungus *P. astrolabium* using high-throughput sequencing technology. These viruses were classified into the *Chrysoviridae*, *Partitiviridae*, *Alphaflexiviridae*, and *Narnaviridae* families, and their impact on host fungal virulence and/or host interactions remains an open and important question. Regarding *Chrysoviridae* viruses, it has been reported that chrysoviruses infect numerous fungi, including *F. graminearum*, *M. oryzae*, and *P. crustosum*, causing phenotypic alterations and reductions in host pathogenicity or fungicide resistance [46,47,48]. Although partitiviruses in *Partitiviridae* family infections are generally asymptomatic, certain partitiviruses have been documented to attenuate fungal pathogenicity. For example, Sclerotinia sclerotiorum partitivirus 1 (SsPV1) induces hypovirulence in *Sclerotinia* spp. and *B. cinerea* [50]. Concerning *Alphaflexiviridae* viruses, studies have reported cross-kingdom transmission and interactions with fungal hosts, yet research on their influence on host growth and pathogenicity remains relatively scarce [58]. *Narnaviridae* viruses typically exhibit cryptic infections in hosts, with biological effects ranging from neutral to mildly detrimental (occasionally symbiotic); they are primarily maintained within host populations through vertical transmission [59,60]. In the *P. astrolabium* strains reported in this study, five strains simultaneously harbored multiple partitiviruses, the *Alphaflexiviridae* virus PaAFV1, and the narnavirus PaNV1. Interestingly, the strain WHG8 also harbors the chrysovirus PaCV1 and exhibited a relatively slow growth on PDA. Therefore, we speculate that the WHG8 infected by multiple viruses may be a promising candidate for investigating virus–*P. astrolabium* interactions and exploring its potential as a biocontrol agent. Due to the complexity of these viruses and their interactions with hosts, the virus-free cured strains and virus-infected strains could be constructed from *P. astrolabium* strains that harbor the viruses to investigate their effects on host pathogenicity, growth rate, and other phenotypes in the future. Integrating techniques such as RNA-Seq and functional analysis will further elucidate the mechanisms underlying hypovirulence, providing new perspectives on the potential microecological roles of these viruses in postharvest grape pathogens.

## 5. Conclusions

In this study, high-throughput sequencing technology was used for identifying mycoviruses from six *P. astrolabium* strains. The results showed that a total of six novel viruses were identified, including four dsRNA viruses and two +ssRNA viruses. Among them, the dsRNA viruses belonged to the *Betachrysovirus* genus in the *Chrysoviridae* family, the *Gammapartitivirus* genus, and an unclassified cluster in the *Partitiviridae* family, respectively. For +ssRNA viruses, single-segment genomic viruses with polycistron encoding were classified into the *Potexvirus* genus in the *Alphaflexiviridae* family, and another one was affiliated with unclassified viruses in the *Narnaviridae* family. In addition, we also obtained partial genome information with non-overlapping and discontinuous sequences of five viruses from four families, including *Discoviridae*, *Peribunyaviridae*, *Botourmiaviridae*, and *Picobirnaviridae*. These results enriched the virus diversity and contributed to the understanding of mycovirus evolution in *P. astrolabium*, as well as providing potential mycovirus resources used for biocontrol and a research system of virus–fungus–plant interactions.

## Figures and Tables

**Figure 1 viruses-17-01053-f001:**
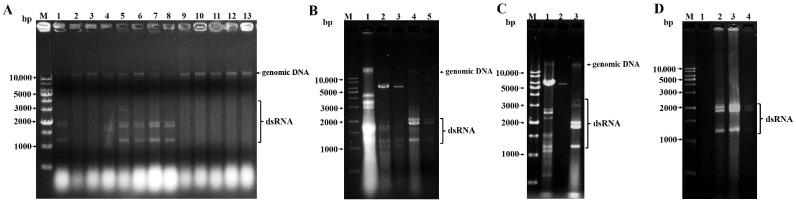
Electrophoretic analysis on a 1% agarose gel for the genomic DNA and the viral genomic dsRNA from *P. astrolabium* harboring viruses. The electrophoretic profile of the total genomic DNA extracted from *P. astrolabium* strains. Lane M: 1 kb ladder marker; Lanes 1–13 correspond to isolates WHG3-3, WHG4, WHG5, WHG6-2, WHG8, WHG9, WHG10, WHG11, CDG12-1, CDG13-1, CDG13-2, CDG15, and CDG16 (**A**). Electrophoretic profile of dsRNA preparations extracted from *Penicillium* spp. Lane M: 1 kb ladder marker (**B**–**D**). Lane 1 corresponds to the isolate CDG6 (another *Penicillium* sp. from grapes). Lanes 2–3 correspond to isolates (another *Penicillium* spp. from garlic) CDS5 and CDS6. Lanes 4 and 5 correspond to *P. astrolabium* isolates WHG3-3 and WHG6-2, respectively (**B**). Lanes 1–2 correspond to isolates (another *Penicillium* sp. from grapes) CDG1 and CDG2-2. Lane 3 corresponds to the *P. astrolabium* isolate WHG8 (**C**). Lane 1 corresponds to the isolate CDG2-1 (another *Penicillium* sp. from grapes). Lanes 2, 3, and 4 correspond to *P. astrolabium* isolates WHG9, WHG10, and WHG11, respectively (**D**).

**Figure 2 viruses-17-01053-f002:**
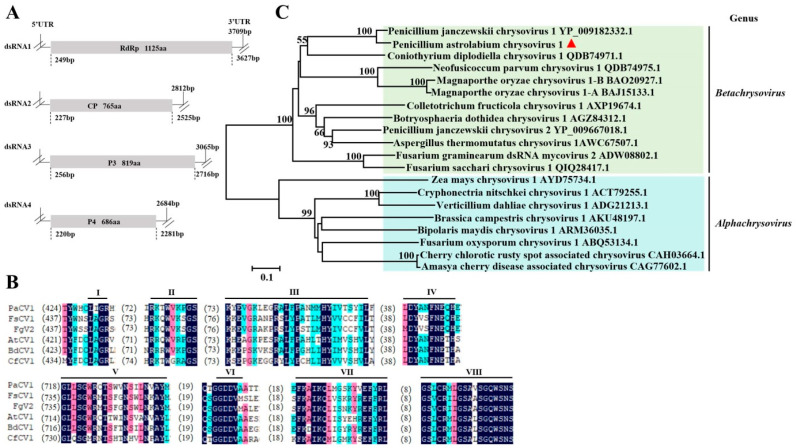
Molecular characterization and evolutionary analysis of PaCV1. Schematic representation of the genomic organization of PaCV1 (**A**). Comparison of the conserved motifs of RdRps encoded by PaCV1 and other selected viruses. The alignment was performed by the program DNAMNA 7.0. Eight conserved motifs of RdRps corresponding to I, II, III, IV, V, VI, VII, and VIII are shown. The numbers within the brackets indicate the number of amino acids not shown in the proteins. FsCV1—Fusarium sacchari chrysovirus 1 (QIQ28417.1); FgV2—Fusarium graminearum dsRNA mycovirus 2 (ADW08802.1); AtCV1—Aspergillus thermomutatus chrysovirus 1 (AWC67507.1); BdCV1—Botryosphaeria dothidea chrysovirus 1 (AGZ84312.1); CfCV1—Colletotrichum fructicola chrysovirus 1 (AXP19674.1) (**B**). Phylogenetic analysis based on PaCV1 RdRp and other selected RdRps. The phylogenetic tree for RdRp sequences was constructed using the neighbor-joining method by MEGA 5.0 with a bootstrapping analysis of 1000 replicates. The virus identified in this work was marked with a red triangle (**C**).

**Figure 3 viruses-17-01053-f003:**
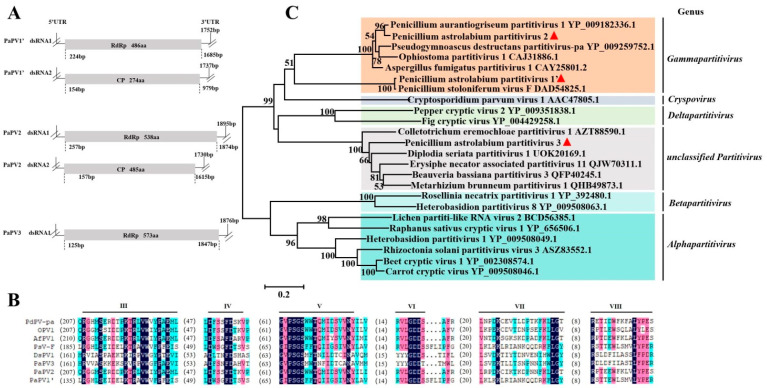
Molecular characterization and evolutionary analysis of PaPV1′, PaPV2, and PaPV3. Schematic representation of the genomic organization of PaPV1′, PaPV2, and PaPV3 (**A**). Comparison of the conserved motifs of RdRps encoded by PaPV1′, PaPV2, PaPV3, and other selected viruses. Six conserved motifs of RdRps corresponding to III, IV, V, VI, VII, and VIII are shown. The numbers within the brackets indicate the number of amino acids not shown in proteins. PdPV-pa—Pseudogymnoascus destructans partitivirus-pa (YP_009259752.1); OPV1—Ophiostoma partitivirus 1 (CAJ31886.1); AfPV1—Aspergillus fumigatus partitivirus 1 (CAY25801.2); PsV-F—Penicillium stoloniferum virus F (AAU95758.1); DsPV1—Diplodia seriata partitivirus 1 (UOK20169.1) (**B**). Phylogenetic analysis of PaPV1′, PaPV2, and PaPV3 RdRps and other selected RdRps. The phylogenetic tree for RdRp sequences was constructed using the neighbor-joining method by MEGA 5.0, with a bootstrapping analysis of 1000 replicates. The viruses identified in this work are marked with a red triangle (**C**).

**Figure 4 viruses-17-01053-f004:**
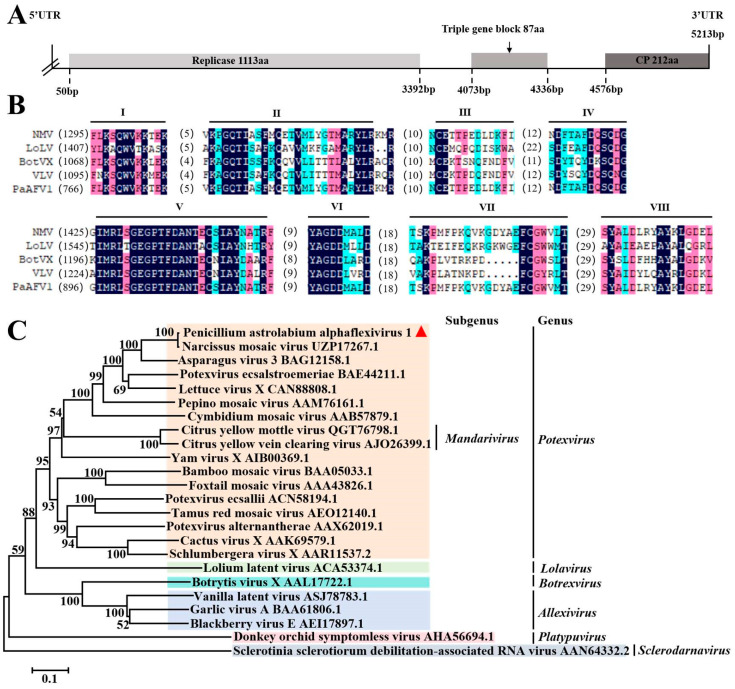
Molecular characterization and evolutionary analysis of PaAFV1. Schematic representation of the genomic organization of PaAFV1 (**A**). Comparison of the conserved motifs of replicases encoded by PaAFV1 and other selected viruses. The alignment was performed by the program DNAMNA 7.0. Eight conserved motifs of Reps corresponding to I, II, III, IV, V, VI, VII, and VIII are shown. The numbers within the brackets indicate the numbers of amino acids not shown in proteins. NMV—Narcissus mosaic virus (UZP17267.1); LoLV—Lolium latent virus (ACA53374.1); BotVX—Botrytis virus X (AAL17722.1); VLV—Vanilla latent virus (ASJ78783.1) (**B**). Phylogenetic analysis of PaAFV1 replicase and other selected replicases. The phylogenetic tree for replicase aa sequences was constructed using the neighbor-joining method using MEGA 5.0, with a bootstrapping analysis of 1000 replicates. The virus identified in this work is marked with a red triangle (**C**).

**Figure 5 viruses-17-01053-f005:**
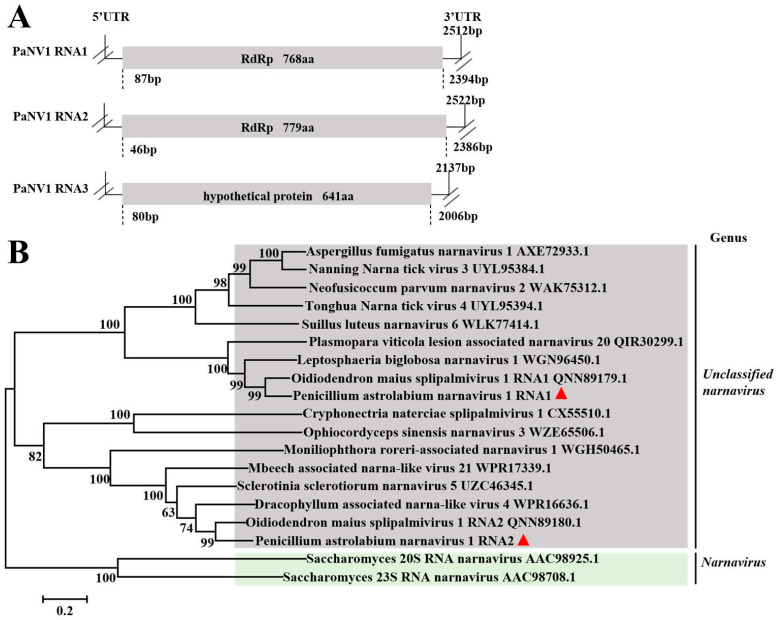
Molecular characterization and evolutionary analysis of PaNV1 RNA1 and PaNV1 RNA2. Schematic representation of the genomic organization of PaNV1 (**A**). Phylogenetic analysis of PaNV1 RNA1 and PaNV1 RNA2 RdRps, as well as other selected RdRps. The phylogenetic tree for RdRp sequences was constructed using the neighbor-joining method by MEGA 5.0, with a bootstrapping analysis of 1000 replicates. The viruses identified in this work were marked with a red triangle (**B**).

## Data Availability

The raw data supporting the conclusions of this article will be made available by the authors on request.

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
