# Peer review of "High-Throughput Sequencing Reveals the Mycoviral Diversity of the Pathogenic Grape Fungus Penicillium astrolabium During Postharvest"

_viruses, 2025, doi:10.3390/v17081053_

Round 1

Reviewer 1 Report

Comments and Suggestions for Authors

This manuscript reports the discovery of diverse mycoviruses in Penicillium astrolabium isolated from postharvest grapes using high-throughput sequencing. The study contributes to the understanding of mycoviral diversity and raises interesting perspectives on potential biocontrol agents. However, several major concerns need to be addressed before the manuscript can be considered for publication.

Major Comments:

  1. The authors state that six out of thirteen P. astrolabiumisolates were found to harbor viruses, as inferred from additional bands observed during agarose gel electrophoresis of extracted DNA. However, Figure 1A only shows results for eight isolates, and lanes 1, 2, 4, and 5 lack clear genomic DNA bands. It is unclear what the remaining lanes represent in Figures 1B–D, aside from those explained in the figure legend. The authors should (i) include gel results for all 13 isolates, (ii) clarify the identity of each lane in all gel images, and (iii) justify why DNA was used to infer virus presence rather than directly extracting and analyzing dsRNA, which is a more conventional approach for detecting mycoviruses.
  2. The authors report the presence of two distinct RdRP segments and assume these represent two separate narnaviruses. However, this is inconsistent with the known genome organization of splipalmiviruses, which are characterized by segmented genomes encoding a split RdRP, with motif A–B on one segment and motif C–D on another. The three identified segments likely belong to a single virus. Phylogenetic analysis should be performed on the concatenated RdRP sequence, not separately for each segment. The authors should revise the corresponding results and discussion accordingly.

Note: Splipalmiviridae is now an officially recognized family by ICTV (https://ictv.global/taxonomy/taxondetails?taxnode_id=202419612&taxon_name=Splipalmiviridae).

Relevant references include:

Chiba Y, Oiki S, Yaguchi T, Urayama S-i, Hagiwara D. Discovery of divided RdRp sequences and a hitherto unknown genomic complexity in fungal viruses. Virus Evol. 2021, 7(1): veaa101.

Jia J, Fu Y, Jiang D, Mu F, Cheng J, Lin Y, Li B, Marzano S-YL, Xie J. Interannual dynamics, diversity and evolution of the virome in Sclerotinia sclerotiorum from a single crop field. Virus Evol. 2021, 7(1): veab032.

  1. The authors describe a viral sequence with >98% identity to Narcissus mosaic virus, a plant virus. While there are precedents for plant viruses infecting fungi, such occurrences remain rare and require strong evidence. I strongly recommend that the authors confirm the presence of this virus in the fungal host via RT-PCR. In our own experience, similar cases often turn out to be sequencing artifacts or sample contamination, as these plant or animal virus sequences were not detectable upon re-extraction and RT-PCR.

Minor Comments:

Line 13: “hypovirulent mycovirus” should be corrected to "hypovirulence-associated mycoviruses".

Lines 203 and 300: Remove the placeholder text "Error! Reference source not found."

Author Response

Thank you very much for taking the time to review the manuscript " High-Throughput Sequencing Reveals Mycoviral Diversities of the Grape-Pathogenic Fungus Penicillium astrolabium during Postharvest " and putting forward valuable constructive suggestions! Your professional advice plays a crucial role in our in-depth understanding of the problem and improving the quality of our articles. We have carefully and comprehensively revised the manuscript based on your review opinions. Specific as follows:

Comments 1. (i) include gel results for all 13 isolates, (ii) clarify the identity of each lane in all gel images, and (iii) justify why DNA was used to infer virus presence rather than directly extracting and analyzing dsRNA, which is a more conventional approach for detecting mycoviruses.

Response 1: (i) We repeated genomic DNA extraction for all 13 isolates which electrophoresis map (Figure 1A) was replaced in revised manuscript. The lanes 1-13 correspond to isolates WHG3-3, WHG4, WHG5, WHG6-2, WHG8, WHG9, WHG10, WHG11, CDG12-1, CDG13-1, CDG13-2, CDG15, and CDG16 in the Figure 1A.

(ii) Regarding to the description of the identity of each lane of Figure 1B-D: due to the difference in growth rate of different strains, the harvest time of mycelia for dsRNA extraction was not synchronized. Therefore, the dsRNA of 6 strains containing virus was extracted in batches in this paper. The few samples extracted at each time were not all strains presented in this paper. The corresponding lanes of the samples of strains other than those in this paper submitted in the manuscript were omitted without notes. The following additions were made as requested by you.

In Figure 1B, the lane 1 was CDG6 (another Penicillium sp. from grape fruits), lanes 2-3 was CDS5 and CDS6 (another Penicillium sp. from garlic), lanes 4-5 was WHG3-3, WHG6-2 respectively. The lane 1-2 was CDG1 and CDG2-2 (another Penicillium sp. from grape fruits), lanes 3 was WHG8 in Figure 1C. The lane 1 was CDG2-1 (another Penicillium sp. from grape fruits), lanes 2-4 was respectively WHG9, WHG10, and WHG11 in Figure 1D.

(iii) The directly extracting and analyzing dsRNA is indeed a more conventional approach for detecting mycoviruses. The purpose of Figure 1A was not to infer the presence of viruses by extracting total DNA, but we found additional genomic nucleic acid fragments when extracting the genome of Penicillium strains isolated from grape fruits, which were suspected to be the genome of RNA viruses. As you stated, Figure 1B-D presented the dsRNA extraction assays of these strains containing additional genomes separately to infer the presence of mycoviruses.

Comments 2. The authors report the presence of two distinct RdRP segments and assume these represent two separate narnaviruses. However, this is inconsistent with the known genome organization of splipalmiviruses, which are characterized by segmented genomes encoding a split RdRP, with motif A–B on one segment and motif C–D on another. The three identified segments likely belong to a single virus. Phylogenetic analysis should be performed on the concatenated RdRp sequence, not separately for each segment. The authors should revise the corresponding results and discussion accordingly.

Response 2: Sincerely thank you for raising deficiencies in previous manuscript. Indeed, as you said, the three identified segments belonged to a single virus. The report the presence of two distinct RdRp segments and assume these represent two separate narnaviruses was inappropriate. We have revised the corresponding results and discussion in the revised manuscript. As for your suggestion that phylogenetic analysis should be performed on the concatenated RdRp sequence, not separately for each segment. We refer to Oidiodendron maius splipalmivirus 1 in the literature with the highest homology to Penicillium astrolabium narnavirus 1 RdRP, we conduct phylogenetic analysis similar to it accordingly, we believe that phylogenetic analysis is also reasonable. (Sutela, S.; Forgia, M.; Vainio, E. J.; Chiapello, M.; Daghino, S.; Vallino, M.; Martino, E.; Girlanda, M.; Perotto, S.; Turina, M. The virome from a collection of endomycorrhizal fungi reveals new viral taxa with unprecedented genome organization. Virus Evol. 2020, 6, veaa076. doi: 10.1093/ve/veaa076.)

Comments 3. The authors describe a viral sequence with >98% identity to Narcissus mosaic virus, a plant virus. While there are precedents for plant viruses infecting fungi, such occurrences remain rare and require strong evidence. I strongly recommend that the authors confirm the presence of this virus in the fungal host via RT-PCR. In our own experience, similar cases often turn out to be sequencing artifacts or sample contamination, as these plant or animal virus sequences were not detectable upon re-extraction and RT-PCR.

Response 3: Thank you for your constructive suggestion. This was indeed an aspect we had not previously considered. Based on your recommendation, total RNA of WHG3-3, WHG8, WHG9, WHG10 and WHG11 strains harboring Penicillium astrolabium alphaflexivirus 1 (PaAFV1) were reverse transcribed into total cDNA. Specific primers PaAFV1 4581 F / PaAFV1 5103 R (523bp length) were designed and a DNA was amplified by this primer pair from total cDNA. The RT-PCR product was consistent with the target band (523 bp) in Figure S1. The result shown that the assembled viral sequence was not sequencing artifacts or sample contamination occurred.

Figure S1. The detection of PaAFV1 by RT-PCR using primer pairs PaAFV1 4581 F / PaAFV1 5103 R. Lane M: DS 5000 marker; Lanes 1-5 correspond to isolates WHG3-3, WHG8, WHG9, WHG10, and WHG11, respectively.

Comments 4. Minor Comments:

Line 13: “hypovirulent mycovirus” should be corrected to "hypovirulence-associated mycoviruses".

Lines 203 and 300: Remove the placeholder text "Error! Reference source not found."

Response 4: for line 13, we have changed " hypovirulent mycovirus " to " hypovirulence-associated mycoviruses"; for lin 209, we have changed "Error! Reference source not found" to "Figure 1"; for line 310 we have changed "Error! Reference source not found" to "Figure 4". 

Reviewer 2 Report

Comments and Suggestions for Authors

The study is significant as it documents mycovirus biodiversity and contributes to the understanding of fungal-mycovirus interactions in grapes. The paper provides a comprehensive report that employs a combination of molecular assays and sequencing technology to identify mycoviruses from 6 strains of P. astrolabium. The clarity and scientific rigor of the study stand out as clear strengths. The authors provide detailed methodologies that provide transparency in the replication of the study findings. The methodology employed in this study is robust, but some adjustments could improve its clarity and applicability. The manuscript is generally clear, but could benefit from greater consistency in formatting, especially when presenting data comparisons and alignments. The interdisciplinary potential of this work is vast, connecting plant pathology, virology, and biological control. This study raises ethical questions about how the presence of mycoviruses in plant pathogenic fungi impacts on reducing symptom severity in plants and the global food production chain, emphasizing the need for holistic agricultural practices. The research could catalyze new studies on the management of mycoviruses in fungi and inspire integrated strategies for controlling plant diseases. In summation, this work bears significant intellectual merit. It extends our understanding of mycovirus impact on a fungi host, drawing essential connections between, plant pathology and global biological control. This paper advances conversations in virology, plant pathology, and global biological control by offering crucial data and innovative methodologies for mycovirus identification and molecular analysis. The findings illuminate pivotal areas for future research, encouraging further discourse on managing of phytopathogenic fungi. The authors are to be commended for their meticulous approach and the novel insights it fosters, contributing robustly to the field. This paper has potential applications and may inspire future studies. It stands as a commendable addition to both the scientific literature and the ongoing conversation on biological control practices.

Author Response

We are deeply grateful for your thorough evaluation and exceptionally insightful comments on our manuscript. Your recognition of the study's "clarity and scientific rigor", "interdisciplinary potential", and "significant intellectual merit" is profoundly encouraging. We have carefully addressed your suggestions to further enhance the manuscript’s impact. Regarding your suggestion:

Comments 1: "The manuscript is generally clear, but could benefit from greater consistency in formatting, especially when presenting data comparisons and alignments".

Response 1: The journal submission document template format was left indent 4.6 cm, and our previous manuscript did neglect the graphic typographic indent, resulting in inconsistent graphics and text formats. In this revision, we have indented all the figures by 4.6 cm to the left. I would like to express our sincere gratitude for your valuable suggestions again.

Reviewer 3 Report

Comments and Suggestions for Authors

Line14, Line 345, Line 375, change ‘pathogenicity’ to ‘virulence’

Line 16, delete ‘a’

Line 20, change ‘to’ to ‘as’

Line 22, Line 26, Line 30, Line 39, Line 42, Line 88, Line 109, Line 111, Line 203, Line 296, Line 297, Line 324, Line 325, Line 326, Line 339, Line 367, Line 374, Line 440, Line 450, Line 458, Line 463, change ‘and’ to ‘, and’

Line 25, delete ‘as’

Line 28, change ‘they’ to ‘which’

Line 55, Line 67, change ‘was’ to ‘is’

Line 72, change ‘can’ to ‘could’

Line 82, change ‘were’ to ‘are’

Line 99, change ‘virus’ to ‘viruses’

Line 111, change ‘classified’ to ‘be classified’

Line 113, delete ‘different’

Line 121, change ‘still’ to ‘are still’

Line 128, change ‘WHG11’ to ‘WHG11 were’

Line 129, change ‘Province’ to ‘province’

Line 132, change ‘days’ to ‘d’

Line 146, change ‘mycelium’ to ‘mycelia’

Line 183, Line 257, Line 325, change ‘BLASTX’ to ‘BLASTx’

Line 222, change ‘3065 bp and 2684 bp, and’ to ‘3065 bp, and 2684 bp,’

Line 224, change ‘exhibits’ to ‘exhibited’

Line 233, change ‘resembles’ to ‘resembled’

Line 258, change ‘has’ to ‘was’

Line 273, change ‘recommend’ to ‘recommended’

Line 290, change ‘and the genome’ to ‘, whose genome’

Line 321, change ‘a RdRp.’ to ‘a RdRp’

Line 346, delete ‘fungi’

Line 352, change ‘can’ to ‘which can’

Line 363, change ‘harbored virus’ to ‘harboring viruses’

Line 364, change ‘virus’ to ‘viruses’

Line 366, change ‘belong’ to ‘belonged’

Line 368, change ‘has’ to ‘had’

Line 373, change ‘infect’ to ‘infected’

Line 376, change ‘needs to’ to ‘or not needs to’

Line 398, change ‘have’ to ‘had’

Line 439, change ‘strains’ to ‘strain’

Line 487, change ‘First Report of Penicillium olsonii Causing Postharvest Fruit Rot of Grape’ to ‘First report of Penicillium olsonii causing postharvest fruit rot of grape’

Line 496, change ‘Mush-room’ to ‘mushroom’

Line 499, change ‘mycoviruses’ to ‘Mycoviruses’

Line 530-531, change ‘White Colony Selection of Virus-Infected Isogenic Recipients Based on a Chrysovirus Isolated’ to ‘white colony selection of virus-infected isogenic recipients based on a chrysovirus isolated’

Line 550, ‘Penicillium stoloniferum’ should not be italicized

Line 555, change ‘reconsideration’ to ‘Reconsideration’

Line 568, change ‘accelerated’ to ‘Accelerated’

Line 574, change ‘molecular’ to ‘Molecular’

Line 586-587, change ‘Characterization of a Chrysovirus Isolated From the Citrus Pathogen Penicillium crustosum and Related Fungicide Resistance Analysis’ to ‘characterization of a chrysovirus isolated from the citrus pathogen Penicillium crustosum and related fungicide resistance analysis’

Line 591-592, change ‘Hypovirulence-Associated Chrysoviruses and Their Host Fusarium Species’ to ‘hypovirulence-associated chrysoviruses and their host Fusarium species’

Comments on the Quality of English Language

The English of this manuscript should be improved to express more clearly the research.

Author Response

We sincerely appreciate your meticulous review and valuable suggestions for improving linguistic precision and formatting consistency. All requested revisions have been carefully implemented as follows:

Comments:Comments and Suggestions for Authors

Line14, Line 345, Line 375, change ‘pathogenicity’ to ‘virulence’

Line 16, delete ‘a’

Line 20, change ‘to’ to ‘as’

Line 22, Line 26, Line 30, Line 39, Line 42, Line 88, Line 109, Line 111, Line 203, Line 296, Line 297, Line 324, Line 325, Line 326, Line 339, Line 367, Line 374, Line 440, Line 450, Line 458, Line 463, change ‘and’ to ‘, and’

Line 25, delete ‘as’

Line 28, change ‘they’ to ‘which’

Line 55, Line 67, change ‘was’ to ‘is’

Line 72, change ‘can’ to ‘could’

Line 82, change ‘were’ to ‘are’

Line 99, change ‘virus’ to ‘viruses’

Line 111, change ‘classified’ to ‘be classified’

Line 113, delete ‘different’

Line 121, change ‘still’ to ‘are still’

Line 128, change ‘WHG11’ to ‘WHG11 were’

Line 129, change ‘Province’ to ‘province’

Line 132, change ‘days’ to ‘d’

Line 146, change ‘mycelium’ to ‘mycelia’

Line 183, Line 257, Line 325, change ‘BLASTX’ to ‘BLASTx’

Line 222, change ‘3065 bp and 2684 bp, and’ to ‘3065 bp, and 2684 bp,’

Line 224, change ‘exhibits’ to ‘exhibited’

Line 233, change ‘resembles’ to ‘resembled’

Line 258, change ‘has’ to ‘was’

Line 273, change ‘recommend’ to ‘recommended’

Line 290, change ‘and the genome’ to ‘, whose genome’

Line 321, change ‘a RdRp.’ to ‘a RdRp’

Line 346, delete ‘fungi’

Line 352, change ‘can’ to ‘which can’

Line 363, change ‘harbored virus’ to ‘harboring viruses’

Line 364, change ‘virus’ to ‘viruses’

Line 366, change ‘belong’ to ‘belonged’

Line 368, change ‘has’ to ‘had’

Line 373, change ‘infect’ to ‘infected’

Line 376, change ‘needs to’ to ‘or not needs to’

Line 398, change ‘have’ to ‘had’

Line 439, change ‘strains’ to ‘strain’

Line 487, change ‘First Report of Penicillium olsonii Causing Postharvest Fruit Rot of Grape’ to ‘First report of Penicillium olsonii causing postharvest fruit rot of grape’

Line 496, change ‘Mush-room’ to ‘mushroom’

Line 499, change ‘mycoviruses’ to ‘Mycoviruses’

Line 530-531, change ‘White Colony Selection of Virus-Infected Isogenic Recipients Based on a Chrysovirus Isolated’ to ‘white colony selection of virus-infected isogenic recipients based on a chrysovirus isolated’

Line 550, ‘Penicillium stoloniferum’ should not be italicized

Line 555, change ‘reconsideration’ to ‘Reconsideration’

Line 568, change ‘accelerated’ to ‘Accelerated’

Line 574, change ‘molecular’ to ‘Molecular’

Line 586-587, change ‘Characterization of a Chrysovirus Isolated From the Citrus Pathogen Penicillium crustosum and Related Fungicide Resistance Analysis’ to ‘characterization of a chrysovirus isolated from the citrus pathogen Penicillium crustosum and related fungicide resistance analysis’

Line 591-592, change ‘Hypovirulence-Associated Chrysoviruses and Their Host Fusarium Species’ to ‘hypovirulence-associated chrysoviruses and their host Fusarium species’

Response:

For Line14, Line 357, Line 388, we have changed ‘pathogenicity’ to ‘virulence’;

On Line 16, we should deleted the ‘a’ by your seuggestion. But this expression was revised as ‘thereby offering a promising biocontrol tool’ by English language editing Prof. Liu.

On Line 19, we have changed ‘to’ to ‘as’;

For Line 22, Line 25, Line 29, Line 38, Line 43, Line 90, Line 111, Line 113, Line 209, Line 306, Line 307, Line 334, Line 335, Line 336, Line 351, Line 380, Line 387, Line 452, Line 462, Line 495, Line 500, we have changed ‘and’ to ‘, and’;

On Line 24, we have deleted the ‘as’;

On Line 27, we have changed ‘they’ to ‘which’;

On Line 57, Line 70, we have changed ‘was’ to ‘is’;

On Line 75, we have changed ‘can’ to ‘could’;

On Line 84, we have changed ‘were’ to ‘are’;

On Line 101, we have changed ‘virus’ to ‘viruses’;

On Line 118, we have changed ‘classified’ to ‘be classified’;

On Line 115, we have deleted ‘different’;

On Line 122, we have changed ‘still’ to ‘are still’;

On Line 134, we have changed ‘WHG11’ to ‘WHG11 were’;

On Line 135, we have changed ‘Province’ to ‘province’;

On Line 138, we have changed ‘days’ to ‘d’;

On Line 152, we have changed ‘mycelium’ to ‘mycelia’;

For Line 189, Line 267, Line 335, we have changed ‘BLASTX’ to ‘BLASTx’;

On Line 232, we have changed ‘3065 bp and 2684 bp, and’ to ‘3065 bp, and 2684 bp,’;

On Line 234, we have changed ‘exhibits’ to ‘exhibited’;

On Line 242, we have changed ‘resembles’ to ‘resembled’;

On Line 267, we have changed ‘has’ to ‘was’;

On Line 283, we have changed ‘recommend’ to ‘recommended’;

On Line 300, we have changed ‘and the genome’ to ‘, whose genome’;

On Line 331, we have changed ‘a RdRp.’ to ‘a RdRp’;

On Line 358, we have deleted the ‘fungi’;

On Line 364, the expression ‘can’ should be ‘which can’ by your suggestion. But this expression was revised as ‘that can be’ by English language editing Prof. Liu.

On Line 376, we have changed ‘harbored virus’ to ‘harboring viruses’;

On Line 377, we have changed ‘virus’ to ‘viruses’;

On Line 379, we have changed ‘belong’ to ‘belonged’;

On Line 380, we have changed ‘has’ to ‘had’;

On Line 386, we have changed ‘infect’ to ‘infected’;

On Line 389, we have changed ‘needs to’ to ‘or not needs to’;

On Line 411, we have changed ‘have’ to ‘had’;

On Line 451, we have changed ‘strains’ to ‘strain’;

On Line 526, we have changed ‘First Report of Penicillium olsonii Causing Postharvest Fruit Rot of Grape’ to ‘First report of Penicillium olsonii causing postharvest fruit rot of grape’;

On Line 535, we have changed ‘Mush-room’ to ‘mushroom’;

On Line 538, we have changed ‘mycoviruses’ to ‘Mycoviruses’;

For Line 569-570, we have changed ‘White Colony Selection of Virus-Infected Isogenic Recipients Based on a Chrysovirus Isolated’ to ‘white colony selection of virus-infected isogenic recipients based on a chrysovirus isolated’;

On Line 589, italics have been corrected;

On Line 594, we have changed ‘reconsideration’ to ‘Reconsideration’;

On Line 607, we have changed ‘accelerated’ to ‘Accelerated’;

On Line 613, we have changed ‘molecular’ to ‘Molecular’;

For Line 625-626, we have changed ‘Characterization of a Chrysovirus Isolated From the Citrus Pathogen Penicillium crustosum and Related Fungicide Resistance Analysis’ to ‘characterization of a chrysovirus isolated from the citrus pathogen Penicillium crustosum and related fungicide resistance analysis’;

For Line 630-631, we have changed ‘Hypovirulence-Associated Chrysoviruses and Their Host Fusarium Species’ to ‘hypovirulence-associated chrysoviruses and their host Fusarium species’;

The above revised content is specifically reflected in red fonts in the manuscript. Finally, thank you again for your very valuable suggestions for quality improvement of this manuscript.

Reviewer 4 Report

Comments and Suggestions for Authors

Given in attachment

Comments on the Quality of English Language

English could be improved.

Author Response

We sincerely appreciate your corrections regarding the language, grammar, and formatting of this manuscript, as well as your valuable suggestions on its structural organization and scholarly depth. Below are detailed responses addressing each of your specific recommendations

Comments 1. Language and Grammar: The manuscript contains numerous grammatical errors and awkward sentence constructions that impede readability.

Response 1: In the Abstract, the description 'the pathogenic fungus Penicillium astrolabium is mostly causing moldy rot in postharvest grape' should be 'is a primary cause of moldy rot in postharvest grapes' by your suggestion. But this sentence was revised as 'hypovirulence-associated mycoviruses can attenuate the virulence of post-harvest grape-rot pathogens, thereby offering a promising biocontrol tool' by English language editing Prof. Liu. In the Introduction, the expression ' a very necessary task for current study 'should be ' an essential objective for biocontrol of postharvest diseases in grapes' by your suggestion. But this sentence was revised as 'developing potent biocontrol strategies against postharvest diseases in grapes has become a pivotal research priority' by Prof. Liu.

For the Recommendation: A thorough revision by a native English speaker or professional editing service is recommended.

The language of our manuscript has been comprehensively improved by Prof. Dr. Qinsong Liu, who obtained his PhD specialized in molecular biology abroad.

Comments 2. Figure Citations and Formatting Errors: Multiple occurrences of “Error! Reference source not found” appear instead of figure or table references, such as in the Results section (lines 203, 300, 399). This hinders the reader’s understanding and should be fixed prior to publication.

Response 2: We appreciate your suggestions regarding the figure citations and formatting errors in this manuscript. We have carefully corrected all such instances throughout the manuscript. For example, in Line 413, we have replaced ‘Error! Reference source not found.’ with ‘Table S1’. All revised locations are now highlighted in red for your review.

Comments 3. Introduction Structure: The introduction provides sufficient background but could benefit from a clearer structure distinguishing:

  • The economic importance of grape postharvest diseases,
  • The role of Penicillium astrolabium,
  • Known mycoviral interactions in fungi,
  • Gaps in current knowledge.

Response 3: (i) We sincerely appreciate your identification of content gaps in the manuscript. Through comprehensive literature review, we determined that even under cold storage conditions, grapes remain vulnerable to postharvest phytopathogens, including blue mold, gray mold, and rhizopus rot, with documented losses approximating 39% of total production yield and 30% of market value. we have supplemented the following critical data in Lines 39-42 of the revised manuscript.

[Jiang, C.; Shi, J.; Liu, Y.; Zhu, C. Inhibition of Aspergillus carbonarius and fungal contamination in table grapes using Bacillus subtilis. Food Control 2014, 35: 41-48. Doi: 10.1016/j.foodcont.2013.06.054.]

(ii) Thank you for this constructive comment that will be of great help to the integrity of the manuscript. It was reported that the morphological characteristics of P. astrolabium were similar to P. olsonii, and symptoms caused by them were also similar in grapes, thereby complicating the differentiation of two postharvest pathogens in grape productio. Nevertheless, we found that P. astrolabium was more prevalent than P. olsonii in rot grape fruits. Suggesting that P. astrolabium is one of the predominant pathogens causing postharvest blue mold in grape. we have incorporated corresponding revisions in Lines 5-51 of the revised manuscript.

(iii) We are grateful for this constructive suggestion, which has significantly strengthened the rigor of our manuscript. It is reported that although the majority of mycoviruses persist asymptomatically within fungal hosts without changing phenotypic traits, a minority can markedly alter the virulence and morphology of their host. The most common manifestations include reduced host growth rate, abnormal pigment deposition and decreased spore production; the resulting attenuation of host pathogenicity is termed hypovirulence. Previous studies demonstrate that, certain mycoviruses have been successfully harnessed for plant disease biocontrol applications. For example, Cryphonectria hypovirus 1 (CHV1) had been used to control chestnut blight caused by Cryphonectria parasitica in Europ, and Rosellinia necatrix megabirnavirus 1 was also applied to control the apple white root rot diseas. Notably, recent studies have revealed that mycovirus could transform pathogenic fungi into beneficial endophytes and activate plant immunit. Sclerotinia sclerotiorum hypovirulence-associated DNA virus 1 (SsHADV-1) isolated from Sclerotinia sclerotiorum could protect oilseed rape from damage caused by a highly virulent strain of S. sclerotioru. The above is reflected in lines 62-82 of revised manuscript.

(iiii) We sincerely appreciate your identification of limitations in the manuscript. Your insightful suggestions have significantly enhanced the coherence of the manuscript. To date, several mycoviruses with biocontrol potential have been reported in Penicillium species, such as Penicillium digitatum virus 1 (PdV1), Penicillium digitatum polymycovirus 1 (PdPMV1), and Penicillium digitatum Narna-like virus 1 (PdNLV1). However, due to the scarcity of viral resources discovered in P. astrolabium, it remains unknown whether any such viruses possess applied value for biocontrol. The above was added in lines 1233-128 of revised manuscript.

Comments 4. Discussion Depth:

The discussion effectively interprets the data but needs to include more critical reflection on:

  • The potential ecological roles of these mycoviruses,
  • Their influence on fungal virulence or host interaction,
  • Their potential applications in biological control.

Recommendation: Add a paragraph speculating on the biological significance of these viruses and future research directions (e.g., functional studies, phenotypic impacts).

Response 4: We sincerely appreciate your insightful suggestions regarding the discussion section. In accordance with your recommendation, we have added a dedicated paragraph in the section of Discussion. The content of this paragraph is as follows:

In this study, 6 novel viruses were identified from the postharvest grape pathogenic fungus P. astrolabium using high-throughput sequencing technology. These viruses were classified into the Chrysoviridae, Partitiviridae, Alphaflexiviridae and Narnaviridae, and their impact on host fungal virulence and/or host interactions remains an open and important question. Regarding Chrysoviridae viruses, it has been reported that chrysoviruses infect numerous fungi, including F. graminearum, M. oryzae, and P. crustosum, causing phenotypic alterations and reductions in host pathogenicity or fungicide resistance[46-48]. Although partitivirus in family Partitiviridae infections are generally asymptomatic, certain partitiviruses have been documented to attenuate fungal pathogenicity. For example, Sclerotinia sclerotiorum partitivirus 1 (SsPV1) induces hypovirulence in Sclerotinia spp. and B. cinerea[50]. Concerning Alphaflexiviridae viruses, studies have reported the cross-kingdom transmission and interactions with fungal hosts, yet research on their influence on host growth and pathogenicity remains relatively scarce [58]. Narnaviridae viruses typically exhibit cryptic infections in hosts, with biological effects ranging from neutral to mildly detrimental (occasionally symbiotic), and are primarily maintained within host populations through vertical transmission[59, 60]. In the P. astrolabium strains reported in this study, 5 strains simultaneously harbored multiple partitiviruses, the Alphaflexiviridae virus PaAFV1, and the narnavirus PaNV1. Interestingly, the strain WHG8 also harbors the chrysovirus PaCV1 and it exhibited relatively slow growth on PDA. Therefore, we speculate that the WHG8 infected by multiple viruses may be a promising candidate for investigating virus-P. astrolabium interactions and exploring its potential as a biocontrol agent. Due to the complexity of these viruses and their interactions with hosts, the virus-free cured strains and virus-infected strains could be constructed from P. astrolabium strains that harbor the viruses to investigate their effects on host pathogenicity, growth rate and other phenotypes in the future. Integrating techniques such as RNA-Seq and functional analysis will further elucidate the mechanisms underlying hypovirulence, providing new perspectives on the potential microecological roles of these viruses in postharvest grape pathogens.

Comments 5: Minor Comments and Technical Suggestions

Response 5: We are grateful for your valuable suggestions regarding linguistic refinements, compositional standards, and technical methodology in our manuscript. The specific revisions implemented are detailed as follows:

For title, we have changed “Mycoviral Diversities” to “Mycoviral Diversity”;

For abstract, we have changed “; for” to “. For”;

For methods, RNA-seq indeed lacked details on replicates in our study. Because the purpose of RNA-seq here was not for differential gene expression analysis. Instead, RNA-seq generating sequences were mapped on genome of host fungi to remove those sequences derived from host fungi. Subsequently, the unmapped sequences (those not aligning to the host fungi genome) were assembled and potentially identified viral sequences that might originate from host fungus harbored viruses. In addition, sequencing depth of every sample was added description “corresponding to approximately 200× coverage” following sequencing raw data on line 175.

For nomenclature, in our manuscript, “PaCV1” is the abbreviation for Penicillium astrolabium chrysovirus 1, while “PaCV-1”and “PaCV1′” are not used. Instead, the abbreviations employed are “PaPV1” and “PaPV1′”. PaPV1′ is the abbreviation for Penicillum astrolabium partitivirus 1′, which facilitates distinction from the abbreviation PaPV1 (Penicillium aurantiogriseum partitivirus 1) already used in previous studies. Thus, there was no inconsistency in virus names in our manuscript.

For figures, In the manuscript, each genomic schematic diagram maintains proportional correspondence to its physical map scale. Consequently, we omitted separate scale bars in these illustrations. This approach aligns with established conventions in virology literature (see the following list), where genomic organization diagrams are routinely presented without explicit scale markers. We therefore consider this rendering method scientifically appropriate.

[Ghabrial, S. A.; Castón, J. R.; Coutts, R. H. A.; Hillman, B. I.; Jiang, D.; Kim, D. H. Moriyama, H. ICTV virus taxonomy profile: Chrysoviridae. Jiang, D 2018, 99, 19-20. doi: 10.1099/jgv.0.001032.

Vainio, E. J.; Chiba, S.; Ghabrial, S. A.; Maiss, E.; Roossinck, M.; Sabanadzovic, S.; Suzuki, N.; Xie, J.; Nibert, M. ICTV virus taxonomy profile: Partitiviridae. J. Gen. Virol. 2018, 99, 17-18. doi: 10.1099/jgv.0.000985.

Kotta-Loizou, I.; Castón, J. R.; Coutts, R. H. A.; Hillman, B. I.; Jiang, D.; Kim, D. H.; Moriyama, H.; Suzuki, N. ICTV virus taxonomy profile: Chrysoviridae. J. Gen. Virol. 2020, 101, 143-144. doi: 10.1099/jgv.0.001383.]

The above all revised content is specifically reflected in red fonts in the revision manuscript.
